

# Enhancing medical image segmentation with a multi-transformer U-Net

Yongping Dan[1], Weishou Jin[1], Xuebin Yue[2] and Zhida Wang[1]

[1] School of Electronic and Information, Zhongyuan University Of Technology, Zhengzhou, Henan, China
[2] Research Organization of Science and Technology, Ritsumeikan University, Kusatsu, Japan

## ABSTRACT

Various segmentation networks based on Swin Transformer have shown promise in medical segmentation tasks. Nonetheless, challenges such as lower accuracy and slower training convergence have persisted. To tackle these issues, we introduce a novel approach that combines the Swin Transformer and Deformable Transformer to enhance overall model performance. We leverage the Swin Transformer's window attention mechanism to capture local feature information and employ the Deformable Transformer to adjust sampling positions dynamically, accelerating model convergence and aligning it more closely with object shapes and sizes. By amalgamating both Transformer modules and incorporating additional skip connections to minimize information loss, our proposed model excels at rapidly and accurately segmenting CT or X-ray lung images. Experimental results demonstrate the remarkable, showcasing the significant prowess of our model. It surpasses the performance of the standalone Swin Transformer's Swin Unet and converges more rapidly under identical conditions, yielding accuracy improvements of 0.7% (resulting in 88.18%) and 2.7% (resulting in 98.01%) on the COVID-19 CT scan lesion segmentation dataset and Chest X-ray Masks and Labels dataset, respectively. This advancement has the potential to aid medical practitioners in early diagnosis and treatment decision-making.

# INTRODUCTION

The continuous evolution of deep learning has unlocked tremendous potential for computer vision technology within the realm of medical image analysis. In the realm of medical image analysis, medical image segmentation stands as a pivotal domain, wielding paramount significance in clinical diagnosis and a wide array of medical applications (*Du et al., 2020*).

Medical image segmentation plays a pivotal role in distinguishing and isolating specific structures and areas of interest within medical images (*Li et al., 2023*). This process aids healthcare professionals in comprehending and scrutinizing anatomical structures, as well as identifying areas affected by lesions or anomalies in these images. The stability and efficiency of medical segmentation, achieved through automated image segmentation technology, are paramount for enhancing the diagnostic and treatment processes. Automating this critical step not only makes clinical diagnoses more efficient but also

Corresponding author
Yongping Dan, 420076822@qq.com

enables doctors to make diagnoses with greater speed and precision, ultimately benefiting patient care (*Sharp et al., 2014*; *Qin et al., 2021*).

Traditional techniques for medical image segmentation, such as edge detection (*Torre & Poggio, 1986*), thresholding (*Mallik et al., 2005*), and region growth (*Adams & Bischof, 1994*), have historically been employed to address basic segmentation tasks. As machine learning advanced, methods based on support vector machines (SVM) and K-means clustering were introduced (*Hartigan & Wong, 1979*), enabling the handling of more intricate images and enhancing segmentation accuracy. However, these approaches fell short of meeting the increasingly demanding performance criteria. Over the past few years, the emergence of deep learning technology has ushered in a revolutionary transformation in the field of medical segmentation. This development has substantially elevated both the accuracy and stability of medical image segmentation.

At this stage, based on deep learning methods, effective feature representation segmentation patterns are learned from a large number of medical images through end-to-end training so as to realize automatic and accurate medical image segmentation (*Zhao et al., 2019*). Currently, deep learning-based medical segmentation mainly favors neural networks with U-shaped structures. In 2015, U-Net was proposed (*Ronneberger, Fischer & Brox, 2015*), and its appearance led to the application of deep learning in medical segmentation. The classical U-structured network consists of an encoder and a decoder and fuses semantic information at different scales through skip connections to realize pixel-level medical segmentation prediction. It uses multiple down-sampling in the encoder part to reduce the spatial resolution and extract the feature information, and then the encoder part fuses the up-sampled feature maps with the skip-connected feature maps, so as to merge the low-resolution semantic information with the high-resolution multi-scale information, and ultimately get accurate segmentation prediction results. The simple and efficient structure of the U-Net attracts researchers to extend this idea, and many researchers have proposed U-Net++ (*Cui, Liu & Huang, 2019*), Residual U-Net (*Zhang, Liu & Wang, 2018*), Dense U-Net (*Wang et al., 2019*), and Attention U-Net (*Oktay et al., 2018*) for 2D medical image segmentation (*Gu et al., 2019*).

Although many CNN-based U-shaped segmentation networks have achieved excellent performance, CNNs still can't fully meet the needs of medical images with high accuracy requirements. In 2017, a Transformer based on the attention mechanism was used in the task of machine translation, and its emergence changed the direction of development in the field of natural language processing (*Vaswani et al., 2017*). Later, researchers tried to use Transformer for semantic segmentation tasks, and methods combining CNN with Transformer appeared, such as TransUNet (*Strudel et al., 2021*; *Chen et al., 2021*). TransUNet combines the powerful global capability of Transformer and the ability of CNN to be sensitive to local image details, which greatly improves segmentation accuracy. In 2020, Vision Transformer (ViT) was proposed (*Dosovitskiy et al., 2020*), which applies Transformer to the field of computer vision and breaks through the limitations of traditional convolution. Since ViT uses a fixed-size global attention mechanism to deal with large-scale images and complex scenes, which incur high computational complexity and memory consumption, researchers have addressed the limitations by improving ViT. To solve the

limitations of ViT, different improvement schemes, such as the Swin Transformer (*Liu et al., 2021*) and Deformable Transformer (*Zhu et al., 2020*) have been developed. Swin Transformer adopts the mechanism of a shifted window to confine attention to the window, which reduces the computational complexity of processing large-scale images. The Deformable Transformer introduces the idea of variable convolution, which allows the network to dynamically adjust the sampling position and shape, thus obtaining the features and structures in the image more flexibly.

This article tackles the challenges posed by low accuracy in medical images, prolonged model training times, and the limitations of single-structure models. We introduce a novel medical segmentation model that combines the capabilities of Swin Transformer and Deformable Transformer. Our approach utilizes the window attention mechanism from the Swin Transformer to capture essential local feature information. Additionally, we leverage the automatic adjustment capabilities of the Deformable Transformer to optimize sampling positions, expediting model convergence.

To enhance the preservation of spatial information, we introduce skip connections and feature fusion mechanisms. Our contributions can be summarized as follows: (1) Combining Swin Transformer and Deformable Transformer: We merge the strengths of Swin Transformer and Deformable Transformer to create a U-shaped structure. This architectural choice significantly accelerates model training. (2) Add skip connections and feature fusion: We augment the model with additional skip connections and feature fusion techniques. These enhancements facilitate the fusion of feature information from different scales, mitigating information loss and elevating prediction accuracy.

# RELATED APPLICATIONS AND RESEARCH

## Applications of medical image segmentation

Medical image segmentation has been a hot branch in the segmentation field and has produced numerous medical image applications.

1. **Organ segmentation:** Medical image segmentation finds one of its most common applications in the realm of organ segmentation (*Gibson et al., 2018*). This crucial task involves delineating distinct organs or tissues within an image, with a focus on vital structures like the heart (including its atria, ventricles, and associated blood vessels), lungs, and brain (covering brain tissues and the intricate network of blood vessels). The significance of organ segmentation lies in its capacity to aid medical professionals at various stages, ranging from initial diagnosis to mid-term treatment decisions and, ultimately, in formulating precise surgical plans.

2. **Lesion area segmentation:** Advancements in detection technology have led to the identification and investigation of an increasing number of medical conditions, including lesions like tumors and skin abnormalities (*Xia, Yin & Zhang, 2019*). Tumor segmentation plays a pivotal role in various aspects of oncology, encompassing lung tumor segmentation (*Jiang et al., 2018*), breast tumor segmentation (*Singh et al., 2020*), brain tumor segmentation (*Wadhwa, Bhardwaj & Verma, 2019*), and more. Precise tumor segmentation serves as a valuable tool for healthcare professionals by assisting

them in pinpointing, quantifying, and assessing tumors. This, in turn, accelerates the process of tumor diagnosis and facilitates the development of effective treatment plans (*Havaei et al., 2017*).

3. **Cell nucleus segmentation:** Within genetic biology and cell biology, the segmentation of cell nuclei holds significant importance (*Lagree et al., 2021*). Achieving precise segmentation and localization of cell nuclei is essential for researchers, as it enables quantitative analysis of these nuclei and facilitates the study of various cellular biological properties.

Segmentation tasks in the medical field encompass a broad spectrum of applications, ranging from bone segmentation to blood vessel segmentation and beyond. These specialized investigations into medical image segmentation have been instrumental in propelling the field of medicine forward. They have enabled the automation and intelligent analysis of medical images, ultimately resulting in heightened efficiency and precision in medical diagnosis and treatment.

## Research on medical image segmentation technology

The swift progress of deep learning has spurred noteworthy breakthroughs in computer vision technology, particularly within the realm of medical image segmentation. Researchers have crafted specialized network models designed for diverse medical image segmentation tasks, leveraging the potency of deep learning to attain remarkable success in this domain.

Traditional methods (*Torre & Poggio, 1986*; *Sharifi, Fathy & Mahmoudi, 2002*; *Pellegrino, Vanzella & Torre, 2004*) mainly apply edge information for medical image processing. Along with the development of deep learning, CNN-based methods (*Kayalibay, Jensen & van der Smagt, 2017*; *Hofbauer, Jalilian & Uhl, 2019*) utilize convolutional neural networks to extract more important information from image blocks and significantly improve segmentation prediction image quality. The emergence of full connectivity has greatly inspired research workers, producing methods such as *Batra et al. (2019)*; *Zhou et al. (2017)* for pixel-level classification tasks. To solve the gradient vanishing, increasing the depth of the network methods such as *Abedalla et al. (2021)*; *Ikechukwu et al. (2021)* further improved the ability to capture both detailed and global information. With the continuous deepening of the network, networks such as KiU-Net (*Valanarasu et al., 2020*), DeepLevel Network (*Wang et al., 2018*), PSPNet (*Zhao et al., 2017*), *etc.* have emerged. The emergence of the attention mechanism brought a boom to the segmentation field, resulting in methods such as *Li et al. (2020)*; *Fu et al. (2019)* that utilize attention to adaptively select important information, enhancing the network's ability to recognize complex information. Despite their excellent performance, the methods based on convolutional neural networks suffer from the disadvantages of high computational cost and poor migration ability. Therefore, it is necessary to study semantic segmentation detection networks that utilize only Transformer.

The Transformer architecture (*Vaswani et al., 2017*) introduced in 2017 has proven to be highly effective in various sequence-to-sequence natural language processing tasks. It also quickly paved the way for innovative approaches to pixel-level classification tasks. The subsequent lightweight Swin Transformer likewise yielded the excellent medical
semantic segmentation Swin UNet (*Cao et al., 2022*). The emergence of the Deformable Transformer brought to the Transformer structure the adaptive deformability to divert attention to important information regions. Vision Transformer and Swin Transformer have the disadvantages of a large number of parameters and slow convergence speed with small sensory fields, respectively. In order to better utilize the resources and reduce the number of parameters while speeding up the convergence of the model, the Swin Transformer and Deformable Transformer are proposed in this article. In this article, we propose a U-shaped segmentation network that mixes both Swin Transformer and Deformable Transformer.

## METHOD

### Overall architecture

As illustrated in Fig. 1, the network architecture proposed in this article comprises an encoder, a decoder, and skip connections. In the encoder, the medical image is partitioned into non-overlapping "tokens" of size $4 \times 4$. Subsequently, a linear embedding layer is applied to transform the image into a sequence of data, allowing it to be processed by the Transformer. This sequence data vector undergoes two key transformations. First, it learns local information using the window self-attention mechanism of the Swin Transformer, enhancing its global awareness through hierarchical window shifting. Second, local detail capture is improved by incorporating deformable convolution from the Deformable Transformer, enabling adaptive feature extraction in local regions. After each Transformer block in the encoder, patch merging is employed to down-sample the data and increase its dimensionality. The decoding process mirrors these operations. The patch-expanding layer is used for up-sampling and dimensionality reduction, and it integrates feature information from neighboring scales into skip connections for fusion. This multi-scale information addition mitigates information loss during the encoding process due to down-sampling, thereby elevating segmentation accuracy. The final patch expanding layer performs a 4-fold up-sampling to restore the sequence features to the original image dimension and resolution. Finally, the resulting image is linearly transformed back to its original pixel-level segmentation prediction form.

### Swin Transformer block

In contrast to ViT, which employs multi-head self-attention (MSA) for feature extraction, Swin Transformer utilizes a more advanced shift window-based MSA (W-MSA and SW-MSA). The shifted window mechanism treats each window as an individual element, thereby enhancing the connections between elements across windows during the shifting process. This approach effectively reduces the computational and storage demands associated with long sequences, making Swin Transformer highly efficient in processing large-sized images. As depicted in Fig. 2, each Swin Transformer module comprises several components, including a window-based multi-head attention module, GELU nonlinear activation function, multi-layer perceptron (MLP), and LayerNorm (LN) layer. These components collectively enable the Swin Transformer to effectively handle large-sized images while optimizing computational efficiency and memory usage.
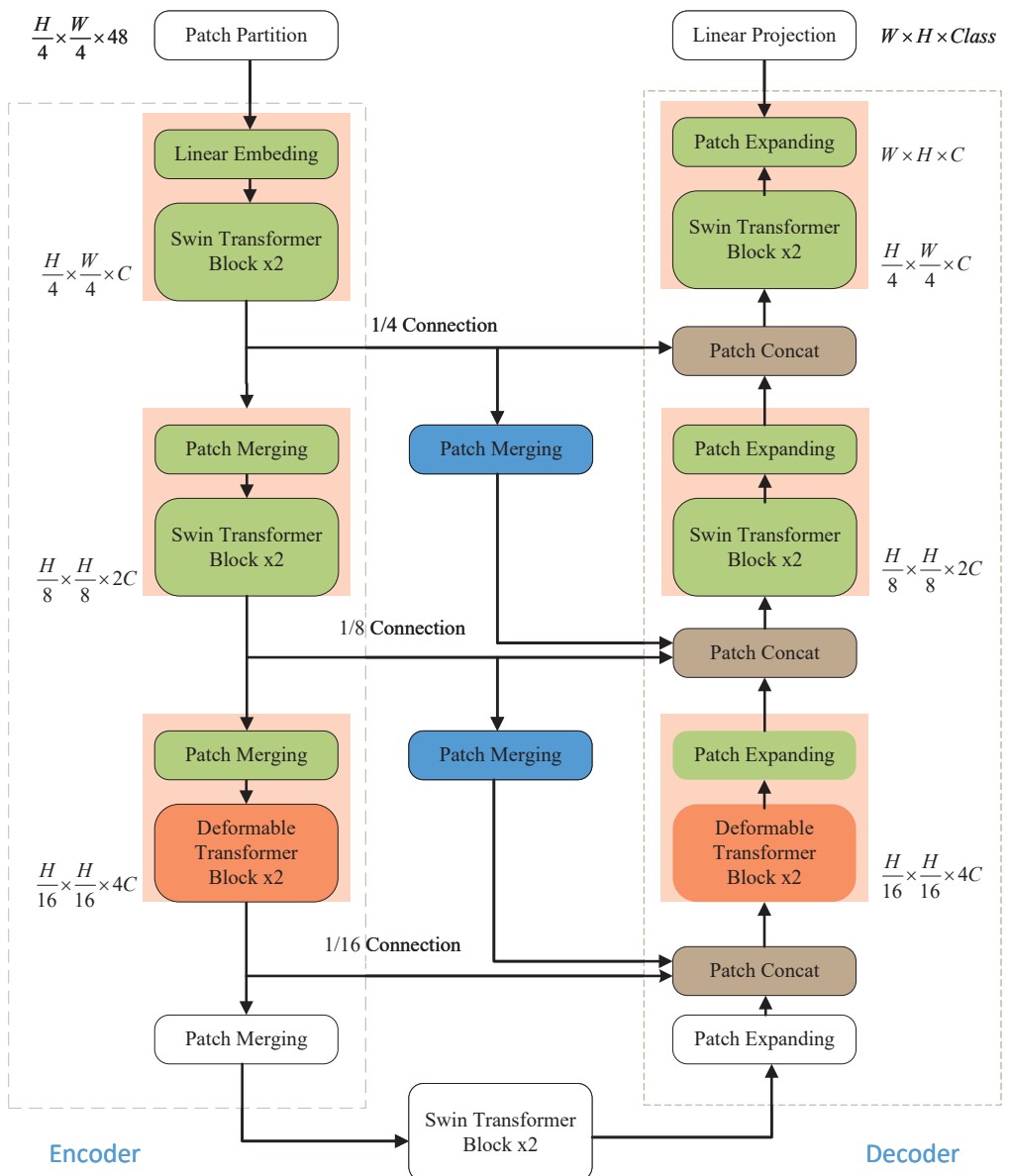

**Figure 1** **Overall structure of the model.** The Swin Transformer and Deformable Transformer serve as the backbone network. Patch merging and patch expanding technologies are employed in the Encoder and Decoder, respectively, to modify the size of feature maps. Furthermore, the model incorporates additional skip connections to enhance multi-scale information fusion, ensuring the retention of crucial information.

The two attention modules (W-MSA and SW-MSA) in Swin Transformer are applied in successive Transformer modules. Based on the shift window mechanism, the two modules are used in different configurations, which can be represented as Swin Transformer modules:

$$\hat{\gamma}^l = W - MSA\left(LN\left(\gamma^{l-1}\right)\right) + \gamma^{l-1} \tag{1}$$

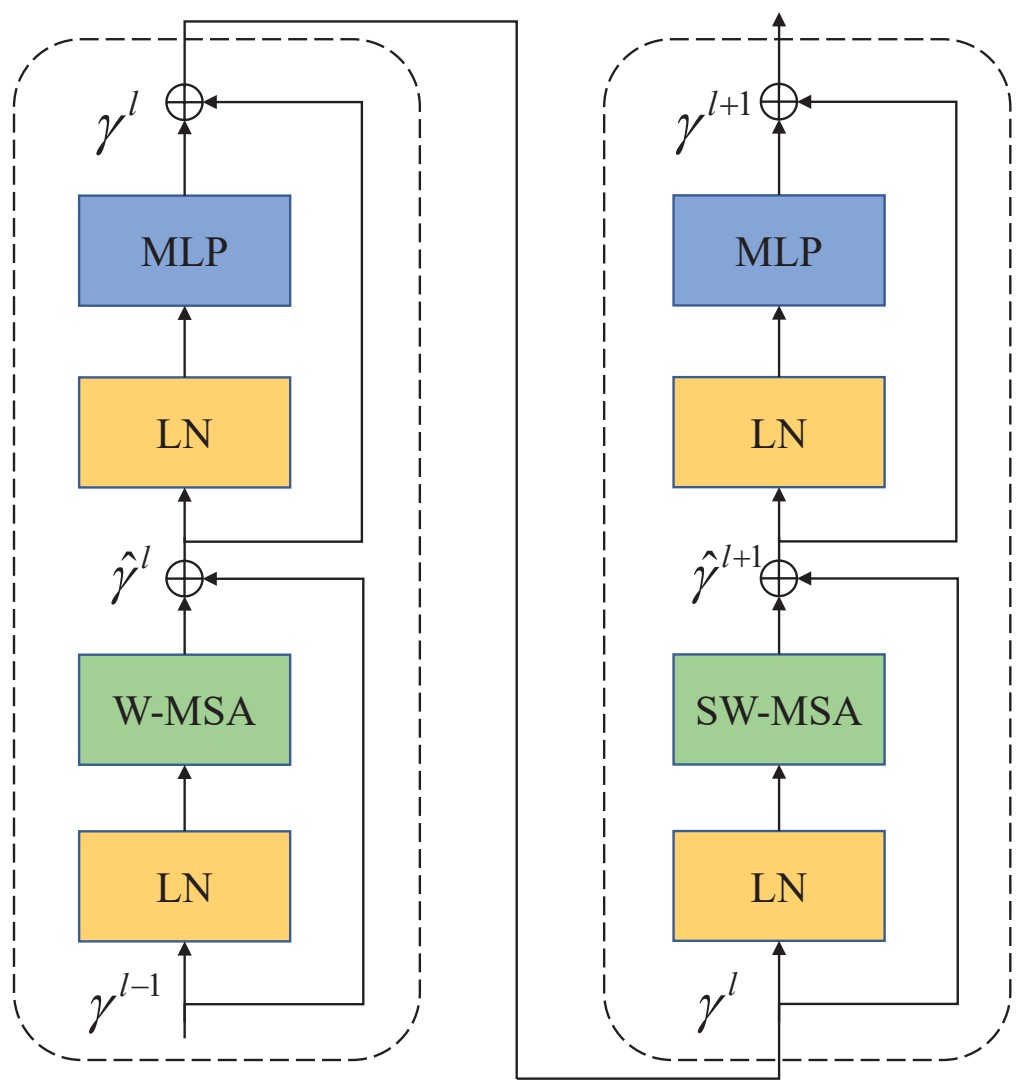

**Figure 2 Swin Transformer block.** The attention blocks with a movable window are composed of W-MSA and SW-MSA attention modules.

$$\gamma^l = MLP\left(LN\left(\hat{\gamma}^l\right)\right) + \hat{\gamma}^l \tag{2}$$

$$\hat{\gamma}^{l+1} = SW - MSA\left(LN\left(\gamma^l\right)\right) + \gamma^l \tag{3}$$

$$\gamma^{l+1} = MLP\left(LN\left(\hat{\gamma}^{l+1}\right)\right) + \hat{\gamma}^{l+1} \tag{4}$$

Similar to the conventional self-attention calculation method, where $\hat{\gamma}^l$ and $\gamma^l$ represent the outputs of the first W-MSA module and MLP module, respectively.

$$\text{Attention}(Q,K,V) = SoftMax\left(\frac{QK^T}{\sqrt{\bar{d}}} + B\right)V \tag{5}$$

where $Q, K, V \in M^2 \times d$ represents query, key, and value matrices. $M^2$ represents the number of patches in a window, and $d$ represents the dimensions of query and key. Since the relative positions of the axes are in [-M+1,M-1], the parameterized bias matrix is $B \in {}^{(2M-1) \times (2M+1)}$.

## Deformable Transformer block

Unlike the Deformable Convolutional Networks (DCNs) in CNNs, which primarily concentrate on more informative regions within the feature map through the deformation of the convolutional filter's receptive field, the Deformable Attention Transformer employs a distinct strategy. It involves learning multiple sets of undisclosed offsets that are used to shift keys and values toward significant regions. Consequently, this approach allows candidate keys and values to be dynamically shifted towards important areas, thereby augmenting the inherent self-attention module with increased adaptability and efficiency in capturing crucial informational features.

As shown in Fig. 3A is the Deformable Transformer Block, which has a similar structure to the Swin Transformer. In Fig. 3B, there is the Deformable attention module. To obtain the offset of the parameter points, the feature input $x \in \mathbb{R}^{H \times W \times C}$ is mapped to the query token $q = xW_q$ and generates the position reference point $p \in \mathbb{R}^{H_G \times W_G \times 2}$. The $q$ is fed into the offset network to generate the offset $\Delta p$. Here is the specific formula provided below:

$$q = xW_q, \bar{k} = \tilde{x}W_k, \tilde{v} = \tilde{x}W_v \tag{6}$$

$$\Delta p = offset(q), \tilde{x} = \phi(x; p + \Delta p). \tag{7}$$

The generated offset $\Delta p$ and the reference position $p$ from the input features make up the position $p + \Delta p$ that generates the displacement. The sampled features $x$ are obtained by adding new displacements to the processing input $\tilde{x}$. Finally, the generated $q, \bar{k}, \tilde{v}$ are fed into the standard multi-head attention to obtain the desired features.

In Fig. 3C, displacement variables are generated by variable convolution. The input feature map size is downsampled by a factor of $r$ $(H_G = H/r, W_G = W/r)$. The values of the parameter positions are constrained to the 2D coordinate system $\{(0,0), \ldots, (H_G-1, H_G-1)\}$, and the mesh shape $H_G \times W_G$ is normalized to a range of $(-1, 1)$, where $(-1, -1)$ denotes the upper left corner and $(1, 1)$ denotes the lower right corner.

The application of deformable attention enables the efficient modeling of token relationships by directing attention to critical regions within the feature graph. These key regions are determined by employing multiple sets of variable sampling points generated by an offset network. Within the offset network, bilinear interpolation is utilized to sample

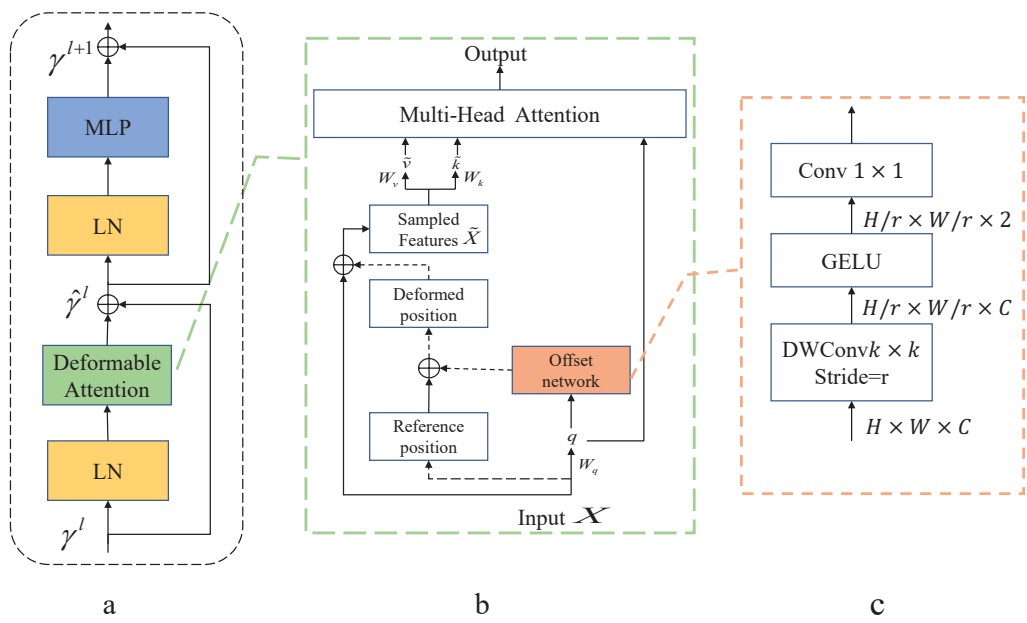

**Figure 3  Deformable Attention block.** (A) This block is structured with a standard attention network architecture. (B) Deformable Attention introduces relative position deviation by incorporating an offset network to enhance the multi-head attention of the output. (C) Provides an overview of the detailed structure of the offset network.

features from the feature map. Subsequently, these sampled feature keys and values undergo projection, yielding the deformed keys and values. Finally, standard multi-head attention is employed to extract and consolidate these deformed features. This approach enhances the learning of deformable attention by introducing robust relative positional bias *via* the positional deformed points.

## Decoder

Just as in the encoder, the decoder undergoes initial processing by the Deformable Transformer block followed by the Swin Transformer block, ensuring symmetry in the final model design. To bring the feature map back to the pixel dimensions of the input image, the encoder employs a patch-expanding layer for up-sampling. Through multiple up-sampling iterations, the feature map is gradually restored to the original image size, ultimately enabling the completion of the segmentation prediction task.

**Patch expanding layer:** After the feature map undergoes a linear layer, doubling its dimensionality, we apply rearrangement and image transformation techniques. These techniques serve the dual purpose of augmenting the size of the image features and reducing the feature dimensionality. Following this, *via* the patch Expanding layer, the feature map is upscaled to a higher-resolution feature map while halving the feature dimension compared to the input dimension.

**Patch concat layer:** To tackle the issue of information loss during the encoding process resulting from consistent downsampling, it is imperative to integrate multi-scale

information fusion. The patch concatenation layer assumes a pivotal role by amalgamating information from the encoder at the same scale, information from neighboring scales, and information from the decoding process. These three components are concatenated and fused through a comprehensive concatenation approach. This fusion mechanism facilitates the exchange of information between the decoding and downsampling phases, effectively bridging shallow and deep features. Consequently, it alleviates spatial information loss attributable to downsampling and contributes to improved segmentation accuracy in the final output.

## Skip connection

The encoding process involves preserving sequence information in a dictionary for later use in the decoding phase. This stored information is then relayed to the decoder, aiding in the recovery of fine details and the integration of contextual information. This approach effectively mitigates the loss of feature information. The skip connection is a crucial component, as it enables the segmentation network to more effectively capture contextual details within the image, ultimately resulting in improved segmentation accuracy in the final output.

## EXPERIMENTS

### Datasets

**COVID-19 CT scan lesion segmentation dataset:** This dataset is COVID-19 lesion masks and their frames from three public datasets (*Morozov et al., 2020*; *Jun et al., 2020*). The dataset contains 2,729 large-size lung CT public datasets with masks. We applied a 9:1 ratio to randomly divide the dataset into training and test sets.

  **Chest Xray Masks and Labels dataset:** This dataset is a public, open segmentation dataset (*Jaeger et al., 2013*; *Candemir et al., 2013*). The dataset contains a training set of 704 X-ray scans and corresponding masks and a training set of five images. To increase the experimental robustness and accuracy, we integrate the dataset and redistribute the training set and test set in a 9:1 ratio.

  The images in datasets have a resolution of $512 \times 512$, and their size is also $512 \times 512$.

### Experiment results on COVID-19 CT scan lesion segmentation dataset

Our proposed model capitalizes on the strengths of Swin Transformer and Deformable Transformer to accelerate convergence and improve prediction accuracy. Given the intricate nature of the self-attention mechanism in Transformers, both training and inference operations necessitate large datasets. In cases where training data is insufficient, performance degradation and poor generalization can occur.To mitigate these challenges, we incorporate the weights of Swin Transformer, which were trained on the ImageNet-1K dataset, into our model using transfer learning. This approach aims to alleviate the potential impact of performance degradation in our Transformer model due to the limited volume of medical dataset data.

  In this experiment, Dice Similarity Coefficient (DSC) and Hausdorff distance (HD), two crucial evaluation indices in medical segmentation networks, are employed.

The Dice Similarity Coefficient (DSC) gauges the similarity between two sets, with a value range of [0, 1]. A higher value signifies greater similarity between the two sets. The calculation involves comparing the enclosed regions in the output prediction image with the original label image to obtain the similarity measure.

$$\text{Dice} = \frac{2TP}{FP + 2TP + FN}. \tag{8}$$

The above formula describes the calculation method for the DSC. In this formula, true positive (TP) represents samples correctly predicted as positive, while the sum of false positive (FP) and false negative (FN) accounts for instances of false alarm and missed detection. The denominator encompasses all correct and incorrect results, and the numerator denotes the accurate number of predicted samples.

HD signifies the maximum distance between the predicted segmentation boundary and the actual region. A smaller HD value indicates a reduced prediction boundary segmentation error and higher quality. Given that the boundary of the reference image is X, the predicted image boundary is Y, and 'd' represents the distance between two points, HD can be expressed by the formula provided.

$$d_H(X, Y) = \max\{d_{XY}, d_{YX}\} = \max\left\{\max_{x \in X} \min_{y \in Y} d(x, y), \max_{y \in Y} \min_{x \in X} d(x, y)\right\}. \tag{9}$$

To mitigate the impact of outlier values and ensure the overall stability of the metric, the actual Hausdorff Distance is determined by selecting the top 95% of distances arranged in ascending order. This approach helps eliminate the influence of extreme values, promoting a more robust assessment of segmentation performance.

As depicted in Fig. 4, our proposed model consistently attains superior segmentation accuracy across various numbers of training epochs. Notably, when the training epochs are limited, our model outperforms the Swin U-Net, which relies solely on the Swin Transformer, by swiftly entering the smoothing convergence phase. Our experiments affirm that our model achieves both higher segmentation accuracy and faster convergence speed.

To verify the validity of our model, we are conducting comparative experiments on different kinds of models.

As shown in Table 1, our proposed model achieves the highest segmentation accuracy; the DSC is 88.18%, which is 1.08% and 0.72% higher than Swing UNet using only Swin Transformer and TransUNet combining Transformer with CNN, respectively. The HD index is the lowest at 9.47. This indicates that our method has better edge prediction.

To demonstrate the effectiveness of the model, we conducted ablation experiments under the COVID-19 CT scan region segmentation dataset. The rigor of the experiment is ensured by conducting control-variable experiments under the condition of specifying a training epoch of 300.

The results of our ablation experiments, as presented in Table 2, unequivocally demonstrate that the inclusion of the Deformable Transformer and skip connections leads to a significant enhancement in final segmentation prediction accuracy. Notably, within the encoder–decoder pipeline, the impact of the Deformable Transformer is
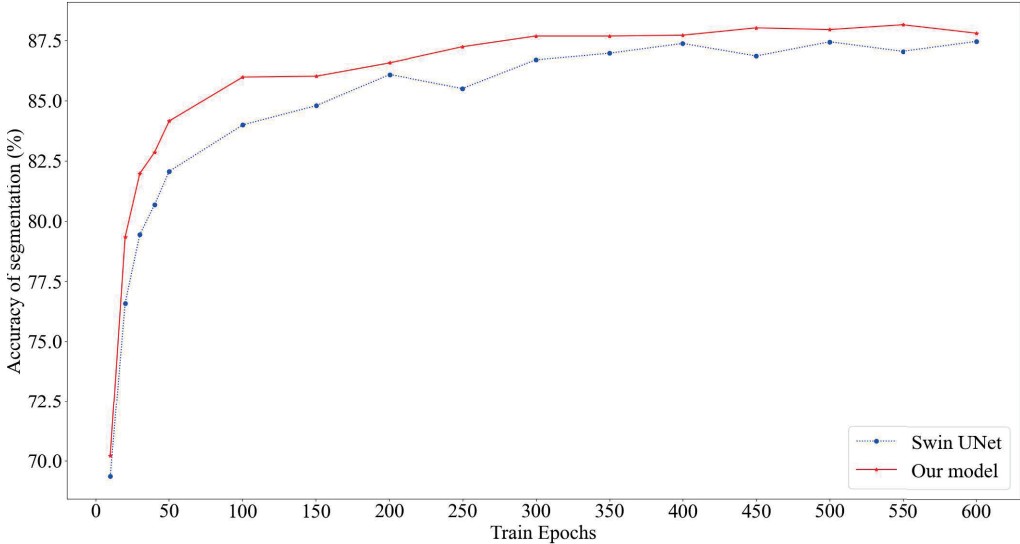

**Figure 4** **Comparison of model segmentation accuracy.** The red section represents the prediction accuracy of our model, while the blue section represents the prediction accuracy of SwinUnet.

**Table 1** **Comparison table of different model experiments.**

| Framework | Average DSC↑ | Average HD↓ |
|---|---|---|
| U-Net(a) | 87.24 | 14.4986 |
| FCN(b) | 86.55 | 16.4849 |
| Deeplab-V3(c) | 86.42 | 15.6495 |
| TransUNet(d) | 87.10 | 11.4432 |
| HiFormer(e) | 86.14 | 17.6833 |
| Swin UNet(f) | 87.46 | 11.5819 |
| Our model | 88.18 | 9.4747 |

Notes.
[a](*Ronneberger, Fischer & Brox, 2015*).
[b](*Dung et al., 2019*).
[c](*Wang et al., 2018*).
[d](*Chen et al., 2021*).
[e](*Heidari et al., 2023*).
[f](*Cao et al., 2022*).
*Average Dice-Similarity Coefficient (DSC); Average Hausdorff Distance(HD).

particularly evident during the decoder phase. It initiates the extraction of feature maps through local attention *via* shift windows, facilitating the collection of localized feature information. Subsequently, deformable attention blocks enhance local tokens by incorporating global relationships into the modeling process. This alternation between local and global attention mechanisms contributes to an enriched model representation. The addition of skip connections supplements the encoding information with that of the decoding process, mitigating the loss incurred by consecutive downsampling, ultimately resulting in improved segmentation accuracy.

**Table 2 Comparative experiments.** The experiments were conducted under the same conditions, and the results were obtained by adding a Deformable Transformer module and a Skip connection at different locations.

| Framework | No deformable transformer | Encoder deformable transformer | Decoder deformable transformer | Encoder+Decoder deformable transformer |
|---|---|---|---|---|
| No add | 86.70 | 87.09 | 87.35 | 87.35 |
| Add 1/4 connection | 87.30 | 87.35 | 87.36 | 87.38 |
| Add 1/8 connection | 87.39 | 87.43 | 87.43 | 87.55 |
| Add 1/4+1/8 connection | 87.40 | 87.46 | 87.59 | 87.69 |

**Notes.**
*This experiment uses the evaluation indicator Average Dice-Similarity Coefficient (DSC).

**Table 3 Different model experiments.**

| Framework | Average DSC↑ | Average HD↓ |
|---|---|---|
| U-Net | 97.75 | 5.1981 |
| FCN | 97.69 | 4.6562 |
| Deeplab-V3 | 97.85 | 4.8562 |
| TransUNet | 97.37 | 5.314 |
| MISSformer | 88.39 | 75.6275 |
| HiFormer | 97.65 | 4.9371 |
| Swin UNet | 95.31 | 4.70 |
| Our model | 98.01 | 4.0674 |

**Notes.**
*Average Dice-Similarity Coefficient (DSC); Average Hausdorff Distance(HD).

## Experiment results on Chest Xray Masks and Labels dataset

To increase the robustness of the experiment, we selected the Chest Xray Masks and Labels dataset for supplementary experiments. In order to solve the problem of randomness and instability caused by a small number of test sets in factor data set planning, we re-plan the training and test ratio of the data set and increase the number of test pictures. Dice-Similarity Coefficient (DSC) and Hausdorff Distance (HD) were used as evaluation indexes.

The results in Table 3 show that our proposed model achieves excellent results under different data sets with an accuracy rate of 98.01%, indicating that the model has good generalization ability and robustness. In addition, as shown in Fig. 5, our proposed model almost perfectly segments lung organs from CT images, and medical workers visually observe the size and shape of the lungs, speeding up the diagnosis and treatment of the disease.

## CONCLUSION

The U-shaped segmentation network we introduced, integrating the features of the Swin Transformer and Deformable Transformer, demonstrates the capability to automatically segment lung tissue from chest CT or X-ray images. This segmentation enables the observation of abnormal organ damage, comprehension of organ location and shape,

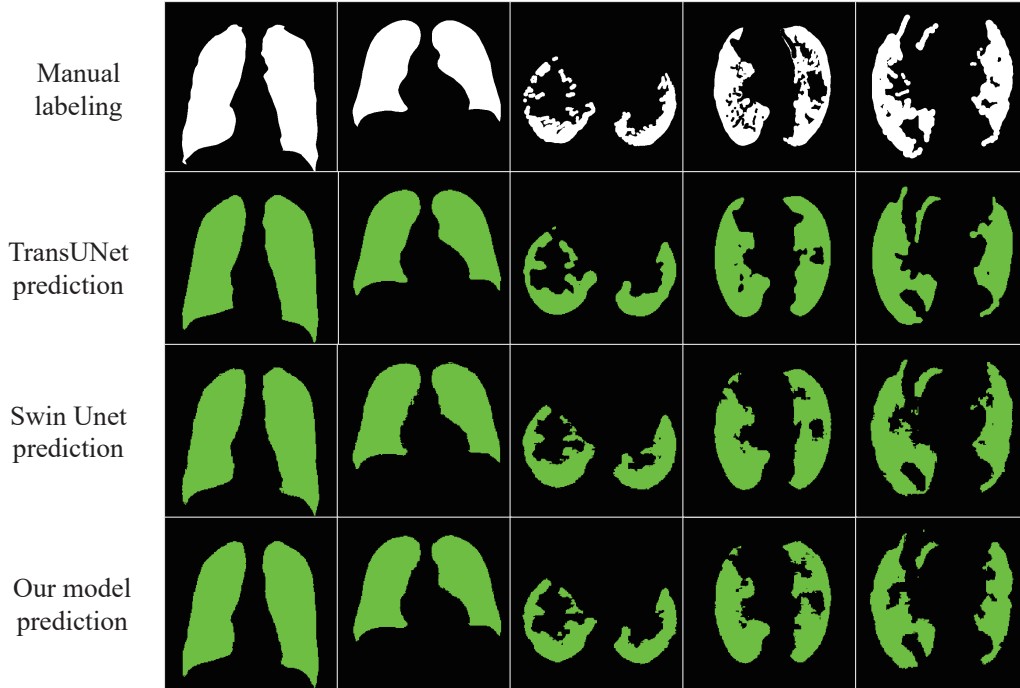

**Figure 5** **Automatic segmentation result.** The lung image is segmented automatically through the network.

and contributes to enhancing the accuracy of diagnosis and treatment. Leveraging the strengths of Transformer blocks and incorporating skip connections for capturing long-range interactive information, our segmentation network exhibits accelerated convergence during training and attains superior prediction accuracy.

Medical image segmentation technology enables automatic organ recognition and lesion detection, the main work of the model at this stage is to optimize for improving the accuracy, and there is not much work on lightweight segmentation models with high accuracy. Therefore, our next stage of research is to lighten the Transformer and reduce the model parameters so that they can be applied to the actual testing equipment.

## ACKNOWLEDGEMENTS

We express our gratitude to the authors of the COVID-19 CT scan lesion segmentation dataset and the Chest X-ray Masks and Labels dataset for generously providing open data sets to support our research endeavors. Furthermore, we have appropriately referenced the articles authored by the creators of these data sets.

### Funding

The authors received no funding for this work.

## Competing Interests

The authors declare there are no competing interests.

## Author Contributions

- Yongping Dan conceived and designed the experiments, analyzed the data, authored or reviewed drafts of the article, and approved the final draft.
- Weishou Jin conceived and designed the experiments, performed the experiments, analyzed the data, prepared figures and/or tables, and approved the final draft.
- Xuebin Yue conceived and designed the experiments, performed the experiments, analyzed the data, authored or reviewed drafts of the article, and approved the final draft.
- Zhida Wang performed the experiments, prepared figures and/or tables, and approved the final draft.

## Data Availability

The code is available in the Supplemental File and at Zenodo: Jin, W. (2023). Data and code [Data set]. Weishou Jin. Available at https://doi.org/10.5281/zenodo.10032283.

The data is available at Kaggle:

- Available at https://www.kaggle.com/datasets/nikhilpandey360/chest-xray-masks-and-labels

- Available at https://www.kaggle.com/datasets/maedemaftouni/covid19-ct-scan-lesion-segmentation-dataset.

## Supplemental Information

Supplemental information for this article can be found online at http://dx.doi.org/10.7717/peerj.17005#supplemental-information.

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
