# Peer review of "Enhancing medical image segmentation with a multi-transformer U-Net"

_PeerJ, doi:10.7717/peerj.17005_

## Round 0.1 · original submission · Major Revisions

When preparing your revised manuscript, you are asked to carefully consider the reviewer comments, and submit a list of responses to the comments. Please take the time to craft detailed responses to the reviewers' questions and comments. If the editors feel that your responses are too brief or have not addressed the reviewers' concerns fully, the manuscript will be returned to you for a more detailed list of responses to the comments.

Reviewer 1 ·

Basic reporting

No comment.

Experimental design

1. What is the hardware used for this experiments? And what is the processing time for this approach?
2. It seems the author used the public datasets to train the model and test the model. The imaging parameters should be included in the method section, i.e. size, resolution of the images.
3. The accuracy measured in figure 4, how exactly is it calculated?
4. How exactly are average DSC and HD measured?
5. Does this proposed model applicable to all the chest CT and X-ray? Please discuss.

Validity of the findings

Although the author provides some detailed description of the proposed models, it seems the final application of this model is just for lung segmentation? And how does segmenting the entire lung be clinical useful? Please provide more discussion on the model's clinical significance. I really don't see this approach offer too much clinical value. I think this is the major weakness for this paper. Simply saying "....,significantly expediting the diagnostic and treatment processes for medical professionals." is too general, the author would need to reference some studies that uses lung segmentation for clinical diagnosis.

Reviewer 2 ·

Basic reporting

The related work part lack of compressive introduction on transformer-based model in medical image segmentation. I recommend the authors refer to the following paper [1]:

[1] Li, Jun, Junyu Chen, Yucheng Tang, Ce Wang, Bennett A. Landman, and S. Kevin Zhou. "Transforming medical imaging with Transformers? A comparative review of key properties, current progresses, and future perspectives." Medical image analysis (2023): 102762.

Experimental design

1. I recommend adding more transformer-based method for comparison as introduced in [1] since right now only 2 transformer-based method is compared compared 3 on CNN-based method.
2. The authors should also report HD for CNN-based method in Table 1,3
3. In experiment results on covid-19 CT scan lesion segmentation task, the used pre-trained weights is unclear and incited.

[1] Li, Jun, Junyu Chen, Yucheng Tang, Ce Wang, Bennett A. Landman, and S. Kevin Zhou. "Transforming medical imaging with Transformers? A comparative review of key properties, current progresses, and future perspectives." Medical image analysis (2023): 102762.

Validity of the findings

1. In Figure 4, the red curve (proposed method) is always below the blue curve (Swin UNet). It looks like the Swin UNet always have superior performance than the proposed method which is contradictory to the conclusion that proposed method have better performance. Also is this figure showing the performance on training set or validation set. It is unclear how the proposed method expedite convergence.
2. Table 2 is hard to follow, I recommend re-organize the table and bold the best result.
3. I recommend adding a qualitatively result on the lesion segmentation task.

Additional comments

1. I suggest including more descriptive details in the caption for both the figures and the tables.

---

## Round 0.2 · Minor Revisions

Please add quantitative results of the lesion segmentation task per Reviewer 2. All other critiques have been adequately addressed.

Reviewer 1 ·

Basic reporting

The revision looks good.

Experimental design

The revision looks good.

Validity of the findings

The revision looks good.

Reviewer 2 ·

Basic reporting

The authors did a good job on answering most of my concerns except for the suggestion on adding a qualitative result on the lesion segmentation task. "Qualitative result" means a visualization of the proposed model results comparing across the different baselines and ground truth, which provide a direct way for the reader to see the model performance.

Experimental design

no comment

Validity of the findings

no comment

Additional comments

I suggest the authors further prove read and check grammar mistake.

---

## Round 0.3 · accepted · Accept

The authors have adequately addressed all critiques and I recommend accepting the manuscript in its current form.